# Resistance to ceftriaxone and penicillin G among contemporary syphilis strains confirmed by natural in vitro mutagenesis

Petra Pospíšilová [1,3], Juraj Bosák [1,3], Matěj Hrala[1], Lenka Krbková[2], Eliška Vrbová[1] & David Šmajs [1] ✉

## Abstract

**Background** For many years, syphilis treatment was considered straightforward due to the universal susceptibility of *Treponema pallidum* subsp. *pallidum* (TPA) to penicillin antibiotics.

**Methods** Penicillin-binding protein genes from a ceftriaxone treatment failure *T. pallidum* isolate were assessed, and the introduction of identified mutations into two laboratory strains via natural competence was aimed for, followed by in vitro analysis of antibiotic susceptibility of the recombinants.

**Results** TPA from the ceftriaxone treatment failure case contained A1873G and G2122A mutations in the TP0705 gene. Introduction of the A1873G mutation into laboratory strains DAL-1 and SS14 resulted in partial resistance to ceftriaxone and penicillin G in vitro. Furthermore, in silico analyses revealed that the majority of contemporary TPA SS14-like strains harbors this mutation and are thus partially resistant to ceftriaxone and penicillin G.

**Conclusions** This finding indicates that TPA strains accumulate mutations that increase their resistance to β-lactam antibiotics. Alternative approaches for controlling syphilis will be needed, including the development of the syphilis vaccine.

## Plain language summary

Penicillin antibiotics have been used to treat syphilis since the 1950s. Resistance to antibiotics is a growing concern. We investigated cases where antibiotics had failed to treat infection and found two mutations in a specific gene that could be responsible. Introduction of one of these mutations into two laboratory *T. pallidum* strains (the bacteria that cause syphilis) resulted in partial resistance to both ceftriaxone and penicillin antibiotics. Moreover, analysis of existing data revealed the presence of this mutation in numerous circulating *T. pallidum* strains, suggesting widespread partial resistance may already exist and increasing concerns about the future emergence of fully resistant syphilis strains.

*Treponema pallidum* subsp. *pallidum* (TPA) is the etiological agent of syphilis, a multi-stage venereal disease that results in 5–7.1 million new infections every year[1]. There are two genetically distinct groups of TPA (i.e., Nichols-like strains and SS14-like strains)[2–4]. The majority of contemporary human TPA isolates belong to the SS14 clade, suggesting its evolutionary advantage over Nichols-like strains[5,6].

Although syphilis diagnosis can be challenging due to variable symptomatology and limitations of current serological tests, syphilis treatment has been considered straightforward, based on the decades-long universal susceptibility of TPA to penicillin[7]. Moreover, similar efficacy for ceftriaxone has been shown in a meta-analysis[8]. Despite the lack of documented clinical evidence of antibiotic resistance, reports of penicillin treatment failure are increasingly numerous[9–11]. In 2017, ceftriaxone treatment failure in congenital syphilis was reported in the Czech Republic[12].

The TPA genome encodes five penicillin-binding proteins (i.e., TP0500, TP0547, TP0574, TP0705, and TP0760 loci). To date, 56 single-nucleotide variants have been described, including 23 in TP0705[13,14].

Recently, a cultivation system for pathogenic treponemes was developed[15], enabling laboratory work with TPA, such as antibiotic susceptibility testing[16–18], and treponemal mutagenesis[19].

In this study, we developed a site-directed mutagenesis approach, utilizing the natural competence of TPA strains, to introduce a mutation identified in TPA from the ceftriaxone treatment failure into the penicillin-binding protein TP0705. The resulting recombinant TPA strains had decreased susceptibility to ceftriaxone and penicillin G in vitro. This is the first experimental evidence of genetic changes conferring partial resistance of TPA to β-lactam antibiotics. Moreover, our findings indicate that the majority of contemporary TPA SS14-like strains have TP0705 alleles encoding this resistance.

## Methods

### Source of *T. pallidum* strains

*T. pallidum* strain SS14 (SS14 clade, allelic profile 1.1.10) was kindly provided by Dr. K. Hawley (University of Connecticut School of

[1]Department of Biology, Faculty of Medicine, Masaryk University, Brno, Czech Republic. [2]Department of Children's Infectious Disease, Faculty of Medicine and University Hospital, Masaryk University, Brno, Czech Republic. [3]These authors contributed equally: Petra Pospíšilová, Juraj Bosák. ✉ e-mail: dsmajs@med.muni.cz

Medicine, Farmington, CT, USA). *T. pallidum* strain DAL-1 (internally marked as PV171; Nichols clade, allelic profile 11.14.10) was kindly provided by Dr. D. Edmondson (University of Texas Health Science Center, Houston, TX, USA). Both TPA strains were provided as frozen suspensions from rabbit testes in glycerol, with unknown concentrations of treponemal cells.

The TPA isolate (SS14 clade, allelic profile 1.26.1), recently described in a case of ceftriaxone treatment failure of congenital syphilis[12], was used for PCR amplification of genes encoding penicillin-binding proteins. This treponemal specimen was isolated in our laboratory and is stored in our laboratory stock.

### Analysis of genes encoding penicillin-binding proteins in TPA isolated from ceftriaxone treatment failure

Analysis of genes encoding penicillin-binding proteins was performed as previously described[20]. Briefly, TP0500, TP0547, TP0574, TP0705, and TP0760 loci were amplified with specific primers, and Sanger sequenced (Eurofins Genomic Europe, Ebersberg, Germany). The nucleotide sequences were compared to sequences present in the SS14 and DAL-1 reference strains[19,21].

### In vitro cultivation of *T. pallidum* strains

Both *T. pallidum* strains (SS14 and DAL-1) were cultivated in vitro over a 1-year period using a previously published cultivation system[15] with some modifications. Briefly, treponemes were cultivated in the presence of TpCM-2 medium (4 ml), Sf1Ep rabbit cells (kindly provided by Dr. D. Edmondson, University of Texas Health Science Center, Houston, TX, USA; 50,000 cells), and a low oxygen atmosphere (2.5%), using a 6-well plate format. Every 7 days, treponemes were collected using Trypsin/EDTA ($2 \times 500$ µl, 37 °C, 5 min) and briefly centrifuged ($150 \times g$, 5 min) to reduce the number of rabbit cells. A supernatant containing treponemes (250–1000 µL) was directly subcultured into a new well containing fresh TpCM-2 medium and Sf1Ep cells.

### Natural in vitro mutagenesis of *T. pallidum* strains

For in vitro mutagenesis, a mixture of five TP0705 PCR amplicons of different lengths was prepared from a clinical sample carrying TP0705 allele from ceftriaxone failure (i.e., an allele containing A1873G and G2122A) using PrimeSTAR GXL DNA polymerase (Takara Bio Europe, Saint Germain en Laye, France). PCR products were purified using the QIAquick PCR Purification Kit (Qiagen) and pooled equimolarly (for more details, see Supplementary Methods).

Long-term in vitro TPA cultures were used as the source of treponemes for mutagenesis. Collected treponemes ($\sim 10^7$ cells) were added to a well containing fresh TpCM-2 medium and Sf1Ep rabbit cells and cultivated as described above. At days 3, 5, and 7, the mixture of TP0705 amplicons carrying 1873G and 2122A (500 ng per addition) was added to the treponemal culture. After 1 week, the medium was replaced with fresh TpCM-2, ceftriaxone at concentration close to the MIC of wild-type strains (2.5 ng/ml, final conc.)[16] was added to fresh TpCM-2 medium, and treponemes were cultivated for another 7 days. Then, treponemes were routinely cultivated in vitro in the presence of ceftriaxone and subcultured (1 ml) every 2 weeks (with a medium exchange in the middle of incubation). Following each passage, the treponemal suspension (5 µl) was immediately subjected to dark-field microscopy to determine treponemal numbers and viability (motility). Another aliquot (200 µl) was frozen (−20 °C) for DNA isolation and subsequent sequence analysis of TP0705 variants.

For monitoring of recombinant TP0705 variants, DNA was isolated using the QIAamp DNA Mini QIAcube Kit (Qiagen) and subsequently used as a template (10 µl) for nested-PCR amplification of the TP0705 fragment as described by Grillová et al.[22]; for more details, see Supplementary Methods. The resulting PCR products were sequenced and analyzed using Geneious Prime (v2024.0). Sequence monitoring of TP0705 was performed every 2 weeks (i.e., for each in vitro subculture) and also during antibiotic sensitivity testing.

### Whole genome sequencing of obtained recombinant TPA strains

The in vitro TPA culture (200 µl) was used for DNA isolation using a QIAamp DNA Blood Mini kit (Qiagen) according to manufacturer's recommendations. For total DNA, the next-generation sequencing (Novaseq X, Illumina, 1GB output) was performed in Novogene (Munich, Germany).

Quality check of the treponemal reads was performed using FastQC (v0.11.5) in conjunction with MultiQC (v1.8)[23,24]. Adapters and low-quality reads were subsequently trimmed using Fastp (v0.20.1)[25]; moreover, reads with a mean quality score below 25 and shorter than 71 nucleotides were discarded. These pre-processed reads were mapped against the genome references of the SS14 strain (GCA_000410555.1_ASM41055v1) and the DAL-1 strain (GCA_000246815.1_ASM24681v1) using the BWA MEM tool (v2.2.1)[26]. The overall mapping quality was assessed using QualiMap (v2.2.2)[27]. Reads that did not align to the reference genome were isolated, and low-quality reads (MAPQ score <60) were filtered out using Samtools (v1.16.1)[28]. Variants were identified using Freebayes (v1.3.6)[29].

### Protein structure prediction

Alphafold structure prediction[30,31] of wild-type penicillin binding protein TP0705 and the version with M625V amino acid change was performed. Wild-type form of TP0705 was downloaded from AlphaFold Protein Structure Database under code A0A0H3BL54. Mutant form of TP0705 newly predicted by AlphaFold and both structures were overlaid in PyMOL (The PyMOL Molecular Graphics System, Version 3.0 Schrödinger, LLC.).

### Susceptibility of TPA to ceftriaxone and penicillin G in vitro

For the determination of antibiotic susceptibility, long-term in vitro TPA cultures were used as a source of treponemes. Using a 24-well plate format, collected treponemes (250,000 cells, determined using dark-field microscopy) were co-cultivated with rabbit Sf1Ep cells (10,000 cells) in TpCM-2 medium (a final volume of 1.5 ml) containing ceftriaxone (Medopharma, Prague, Czech Republic; concentrations of 5.0, 2.5, 1.25, and 0.0 ng/ml) or penicillin G potassium salt (Biotika, Slovenská Lupča, Slovakia; concentrations of 0.5, 0.25, 0.125, and 0.0 ng/ml). The concentration ranges were derived from preliminary experiments determining the concentrations inhibiting 90% of wild-type TPA culture. After 7 days of cultivation under a low-oxygen atmosphere (2.5%), treponemes were collected using Trypsin/EDTA treatment ($2 \times 250$ µl). Cell suspensions of treponemes (5 µl) were immediately subjected to dark-field microscopy for analysis of treponemal viability (motility). Another aliquot ($2 \times 5$ µl) was frozen (−20 °C) and used as a template for qPCR analysis of treponemal numbers.

qPCR detection was based on the detection of the *pol*A gene (TP0105; product size 129 bp) using probe qPCR_polA_probe (5'–FAM-TCCGCTT GGAAACAGCAGGATTG-BHQ–3') and the Azure Cielo Real-Time PCR System (Azure Biosystems), as described previously[32] (for more details, see Supplementary Methods). At least three biological replicates were performed for each antibiotic concentration, and each replicate was tested in duplicate using qPCR analysis. There were 10 biological replicates (20 values) for the SS14 strain, 6 biological replicates (12 values) for the recombinant SS14 A1873G strain, 5 biological replicates (10 values) for the DAL-1 strain, and 3 biological replicates (6 values) for the recombinant DAL-1 A1873G strain in ceftriaxone testing, and 20, 10, 12, and 6 values, respectively, in penicillin G testing.

### Statistical analysis

A Mann–Whitney test was used for statistical analyses of treated/untreated ratio. *p* values less than 0.05 were considered statistically significant and are denoted with asterisks according to statistical significance (∗*p* < 0.05, ∗∗*p* < 0.01, and ∗∗∗*p* < 0.001). Secondary minimal inhibitory concentration (MIC) value was defined as the lowest antibiotic dilution where the TPA numbers were significantly lower than non-antibiotic control, as described previously[16]. To determine secondary MIC comparison, an absolute numbers of copies revealed by qPCR were used, and statistical significance was supported by Kruskal–Wallis mean rank test followed by Dunn´s test for

**Fig. 1 | Scheme of the natural in vitro mutagenesis of the TPA SS14 strain.** The mixture of TP0705 PCR products, containing A1873G and G2122A mutations present in TPA from a ceftriaxone failure case, was added to an in vitro culture of the SS14 strain. From day 7, ceftriaxone (2.5 ng/ml of culture medium) was used for the selection of recombinants. At day 140, the A1873G mutation was fixed in the SS14 culture. At day 212, this mutation was not detected in the control treponemal cultures without the PCR mixture and ceftriaxone. The treponemal culture without the PCR mixture was eliminated in the ceftriaxone-supplemented control. The arrows indicate position 1873 of the TP0705 gene. As G2122A recombinants were not found during the experiment, this position is not shown. The scheme contains real sequences from the experiment at relevant time points after mutagenesis.

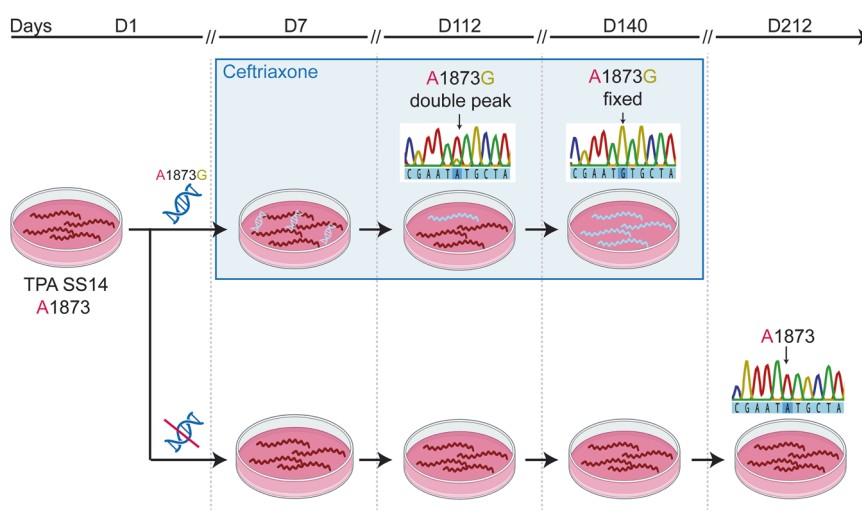

pairwise comparisons between specific antibiotic concentration and non-antibiotic control group. GraphPad Prism 10 software was used for calculations.

### Declaration of Generative AI and AI-assisted technologies in the writing process
During the preparation of this work no AI was used.

### Reporting summary
Further information on research design is available in the Nature Portfolio Reporting Summary linked to this article.

## Results
### TPA from the ceftriaxone treatment failure harbored two nucleotide variants in the TP0705 gene encoding penicillin-binding protein
In the Czech Republic, ceftriaxone treatment failure was recently reported in a child with congenital syphilis (SS14 clade, allelic profile 1.26.1)[12]. An infant with serologically confirmed syphilis received intravenous ceftriaxone (100 mg/kg daily for 14 days). Five months later, follow-up serology showed an elevated RPR titer, and a buccal swab revealed an identical allelic profile to that observed before ceftriaxone therapy. The reinfection was excluded by testing of parents for active infection, and intramuscular procaine benzyl-penicillin (50,000 units/kg daily for 14 days) was subsequently administered.

Nucleotide sequences of five penicillin-binding proteins were analyzed for this TPA isolate. While TP0500, TP0547, TP0574, and TP0760 sequences were identical to the SS14 reference strain[33], the TP0705 sequence revealed two nucleotide variants (A1873G and G2122A substitutions) compared to the SS14 strain. These variants resulted in amino acid changes M625V and G708S, respectively.

### Natural in vitro mutagenesis resulted in the introduction of the A1873G mutation into laboratory TPA strains
The TP0705 gene of two laboratory strains was mutagenized using a PCR mixture sequentially corresponding to the TP0705 allele present in TPA from the ceftriaxone failure case (i.e., 1873G and 2122A) via natural mutagenesis (Material and Methods). In vitro cultivation of TPA with ceftriaxone resulted in a detectable presence of the recombinant variant of the TP0705 gene. Interestingly, only the A1873G mutation was selected, despite the PCR mixture contained both the A1873G and G2122A mutations. In the SS14 strain, the A1873G mutation was fixed at day 140 (Fig. 1). During an additional 210 days of in vitro cultivation without ceftriaxone, the fixed A1873G mutation failed to revert. The same mutation was also fixed in TPA strain DAL-1 belonging to the Nichols clade (for more details, see

Supplementary Fig. S1a). In vitro growth of *T. pallidum* during the selection of recombinant strains SS14 A1873G and DAL-1 A1873G is depicted in Supplementary Fig. S2.

All relevant controls were negative for TP0705 recombinants. Furthermore, whole genome sequencing of both recombinant strains was performed and compared to wild-type SS14 and DAL-1 strain sequences. The recombinant strains were shown to carry A1873G sequence change in TP0705 (read counts supporting this change were 923/923 and 1072/1080 in SS14 and DAL-1 recombinant strains, respectively), and no other sequence changes common to both strains were detected. While the DAL-1 recombinant strain harbored only A1873G sequence change, two additional SNPs were detected in the SS14 recombinant strain, including C200A in TP0732 (genome coordinate G799578T, i.e., amino acid change T67K in bifunctional methylenetetrahydrofolate dehydrogenase (NADP(+))/methenyltetrahydrofolate cyclohydrolase) and G1273A in TP0746 (genome coordinate C811984T, i.e., amino acid change A425T in pyruvate, phosphate dikinase) (genome coordinates and annotation based on CP004011). Because it is unlikely that these genes are relevant for susceptibility to β-lactams, both recombinant strains were assessed for testing of antibiotic susceptibility.

### Recombinant treponemes in TP0705 (A1873G) were partially resistant to ceftriaxone and penicillin G
The recombinant strains (A1873G), as well as the original wild-type strains (i.e., isogenic controls), were assessed for susceptibility to ceftriaxone and penicillin G, which are used in clinical practice. Although AlphaFold prediction did not reveal a prominent change of protein structure related with introduced mutation in TP0705 (Supplementary Fig. S3), an increased resistance of the recombinant SS14 strain was statistically confirmed in all tested concentrations (Fig. 2). In the case of ceftriaxone, the A1873G mutation affected TPA numbers as well as treponemal motility (see Fig. 2a, ceftriaxone conc. 2.5 ng/ml).

Introduction of the A1873G mutation into the DAL-1 strain had a comparable effect on ceftriaxone and penicillin G susceptibility as that seen in the SS14 strain (Supplementary Fig. S1b).

Moreover, in accordance with Tantalo[16], the secondary minimal inhibitory concentration (MIC) was determined for the recombinant strains. For ceftriaxone, both recombinant strains showed a two-fold higher secondary MIC concentration compared to wild-type strains (from 2.5 to 5.0 ng/ml). Similarly, both mutants showed a higher secondary MIC for penicillin G. The MIC of the DAL-1 mutant increased from 0.25 to 0.5 ng/ml, while the trend for increased resistance was even higher for SS14 mutant ( > 0.5 ng/ml). For more details, see Supplementary Fig. S4.

**Fig. 2 | Susceptibility of the recombinant SS14 strain to ceftriaxone and penicillin G determined under in vitro conditions. a** Susceptibility of the recombinant SS14 strain to ceftriaxone. **b** Susceptibility of the recombinant SS14 strain to penicillin G. The introduction of the A1873G mutation resulted in decreased susceptibility to both tested antibiotics. The graphs show the numbers of *pol*A genes detected in the in vitro culture (at day 7, using qPCR). All detected *pol*A numbers were normalized to the numbers present in the corresponding controls without antibiotics. The red line represents the median ratio value, the boxplot shows IQR, and the colored dots represent individual ratio values. At least three biological replicates and two technical replicates were examined using qPCR and are shown. Motility data were obtained using dark-field microscopy. The Mann–Whitney test was used to calculate the statistical significance (*$p < 0.05$, **$p < 0.01$, and ***$p < 0.001$).

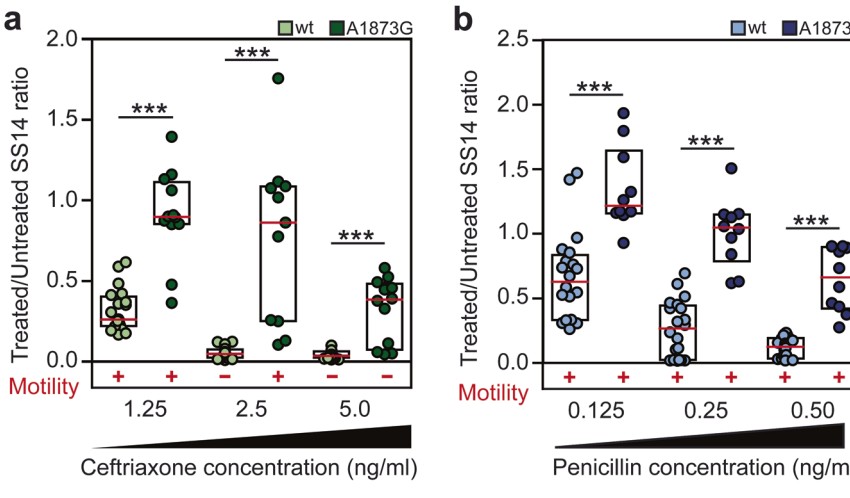

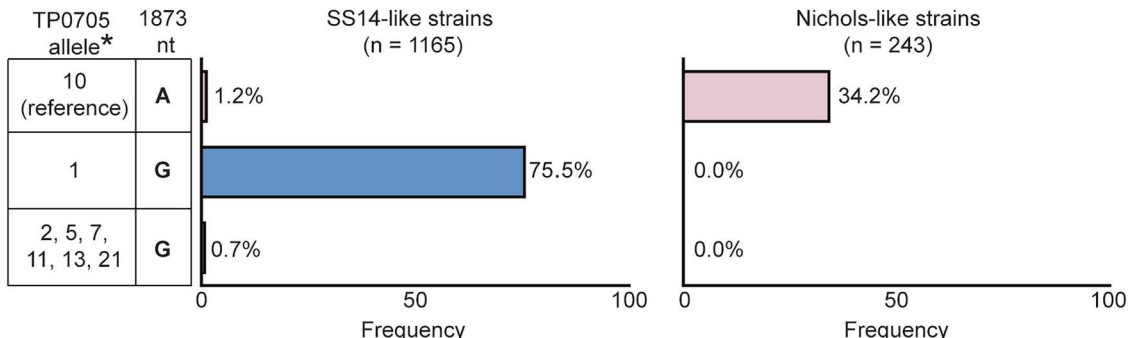

**Fig. 3 | In silico analysis of TPA sequences containing the 1873G variant at the TP0705 locus responsible for partial resistance to the tested antibiotics.** Data were taken from the PubMLST database[14] and included 1408 isolates. Allele 10 is considered the reference allele since it was found in both reference TPA strains-Nichols and SS14. Allele 1 is the most common among contemporary SS14-like strains. *Among the rare 1873G alleles, alleles 4 and 6 (*n* = 2 isolates) were omitted from the analysis due to the absence of SS14/Nichols clade classification.

## In silico analysis of the TP0705 locus revealed a predominance of alleles having the A1873G variant among contemporary SS14-like isolates

Based on multi-locus sequence typing[22], the TPA allelic profile represents a combination of allele variants from three genomic loci (TP0136, TP0548, and TP0705). Analysis of TPA isolate sequences deposited in the PubMLST database[14] revealed that a considerable number of TPA isolates (61%; 951 out of 1559) carry the 1873G nucleotide variant. Among the twenty-one known TP0705 alleles, this nucleotide variant was found in alleles 1, 2, 4, 5, 6, 7, 11, 13, and 21. The recombinant strains, experimentally prepared by ceftriaxone selection in this study, correspond to TP0705 allele 21; other eight alleles have 1873G in combination with additional TP0705 nucleotide variants.

For available SS14- and Nichols clade classification (*n* = 1408 TPA isolates), the 1873G variant was found only in the SS14 clade, where it represents 76.2% of isolates (Fig. 3). While alleles 2, 5, 7, 11, 13, and 21 were found rarely (isolates of alleles 4 and 6 were not classified to TPA clade as these two alleles were found in a single isolate each and TP0705 was the only locus determined), allele 1 was predominant (75.5%, 880 out of 1165 SS14-like isolates).

## Discussion

In this study, we demonstrated that the A1873G substitution within the TP0705 gene, experimentally introduced into SS14 and DAL-1 strains using an in vitro cultivation system, confers resistance to ceftriaxone, penicillin G, and potentially other β-lactam antibiotics in contemporary SS14-like strains

of TPA. The TP0705 is a gene that encodes a penicillin-binding protein. It shares sequentially homology with penicillin-binding protein 1 (PonA) found in *Neisseria gonorrhoeae*, which is known to mediate resistance to β-lactam antibiotics through the emergence of a single mutation[34]. However, the sequence homology between protein TP0705 and PonA is insufficient to infer further similarities. As is known from many other bacterial pathogens[35,36], the stepwise acquisition of antibiotic resistance through the accumulation of point mutations, either within the same or functionally related proteins, can result in a substantial level of resistance. Our discovery of the A1873G substitution in the TP0705 gene of contemporary TPA strains could very well be a step in this process.

The mentioned mutation resulted in the amino acid change M625V, located within an α-helix, and did not prominently alter the secondary structure of the whole protein (AlphaFold structure prediction[30,31], Supplementary Fig. S3), suggesting that M625V replacement could modify the interaction with the penicillin molecule. Although we observe no major differences, an impact on penicillin binding cannot be excluded.

Historically, the absence of reports describing penicillin resistance in TPA appeared to answer the question of whether TPA has the genetic capacity to develop antibiotic resistance to penicillin[7]. Despite numerous reports of failed syphilis treatment[9–12], resistance to β-lactam antibiotics remained unproven, further delaying investigation of this question. In this study, the TPA cultivation system[13] allowed us to address this issue experimentally and demonstrate that TPA is genetically capable of encoding penicillin resistance, albeit to a limited extent that still allows the clinical use of β-lactam antibiotics. The drug plasma concentration of ceftriaxone

(29.7 mg/L[16]) appears sufficient to overcome the decreased susceptibility of the mutant TPA strain (0.0025 mg/L for SS14 strain and 0.005 mg/L for SS14 A1873G mutant). The drug plasma concentration of benzathine penicillin G (0.0125 mg/L[16]) should also be sufficient, as secondary MIC assessed in our study for A1873G mutant is higher than 0.0005 mg/L, which is at least 4 times higher than for the wild-type SS14 strain. A limitation of this approximation is the difference between benzathine penicillin G, used for syphilis treatment, and penicillin G potassium salt, used in this study. However, relative in vitro growth inhibition of these two penicillin forms has been shown to be similar[17]. Nevertheless, tissue concentration of the drug may differ from plasma concentration, and thus, A1873G mutants should be tested in experimental rabbit infection to address whether therapeutic levels of penicillin are sufficient.

Regarding the use of macrolide antibiotics in syphilis treatment, the emergence of two mutations in 23S rRNA (both copies) was described in the early 2000s[37,38]. Within two decades, macrolide-resistant TPA strains had spread to most developed countries, where they now account for over 80% of circulating TPA strains[39]. Furthermore, similar mutations have recently emerged following macrolide treatment of yaws (*T. pallidum* subsp. *pertenue*)[40–43] and endemic syphilis (*T. pallidum* subsp. *endemicum*)[44], thus repeating the process that already took place in TPA isolates two decades ago. These findings suggest that the emergence and spread of antibiotic resistance in TPA is a relatively common process. We, therefore, speculate that the emergence of clinically relevant penicillin resistance in syphilis is possible and could occur in a relatively near future.

In this study, we developed a treponemal site-directed mutagenesis protocol based on the uptake of homologous linear DNA (a PCR product) by acceptor treponemal cells. Pathogenic treponemes are thus naturally competent (i.e., capable of DNA uptake), likely a result of DNA transfer during mixed infections, as previously predicted by studies demonstrating inter-subspecies recombination in TPA[45–47]. Experimental de novo emergence of the A1873G mutation in a control treponemal culture was not observed, which is consistent with the relatively low treponemal evolution rates (estimated at $2.8–4.1 \times 10^{-10}$ nucleotide changes per site per generation)[48,49] and the limited number of cells (~$10^7$) used in our experiment. In general, transformation mediated by natural competence appears to be more frequent than point mutations by at least several orders of magnitude. This molecular mechanism could play an important role in treponemal evolution.

TP0705 allele 1, harboring A1873G and G2122A variants found in TPA from ceftriaxone treatment failure, has been found in 75% of the SS14-like isolates investigated to date. Furthermore, as over 80% of circulating syphilis strains are SS14-like strains[5,14], this suggests that well over half of TPA strains are partially resistant to ceftriaxone, penicillin G, and potentially other β-lactam antibiotics. While we were able to experimentally introduce A1873G mutation into the DAL-1 strain, no Nichols-like isolates harboring this mutation have been identified thus far. The partial resistance to penicillin observed in contemporary SS14-like strains of TPA could contribute to the paradox of Nichols-like strains[5,50], which are common among historical laboratory reference strains but less common among contemporary clinical isolates, particularly in developed countries. Recent reports show that Nichols-like strains are more prevalent than previously analyzed[51,52]. A very recent report found an almost 1:1 ratio between SS14 and Nichols clades in a set of 96 typeable clinical samples collected in Spain between 2021 and 2023[53], and it will certainly be interesting to observe how the prevalence of SS14- and Nichols-like isolates will be changing over time in different locations.

Most importantly, the existence of the mutation responsible for partial resistance to ceftriaxone and penicillin G has been confirmed, and its broad presence among contemporary syphilis isolates suggests ongoing spread of this resistance. Moreover, there is a potential risk that the emergence and accumulation of other mutations will further increase the degree of TPA resistance to penicillin and related β-lactam antibiotics. Therefore, alternative approaches to control syphilis infection, including the development of a syphilis vaccine, will be needed in the future.

## Data availability

Source data for Figs. 2, S1 and S4 can be found in Supplementary Data 1. Sequencing data were submitted to the NCBI BioProject database under the following accession numbers PRJNA1232603 (SS14 strain carrying A1873G) and PRJNA1232606 (DAL-1 strain carrying A1873G).

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

## Acknowledgements

We thank Thomas Secrest (Secrest Editing, Ltd.) for the English editing of the manuscript. Figures were partially created by Biorender.com. D.Š. discloses support for the research of this work from the National Institute of Virology and Bacteriology project (Programme EXCELES, ID Project No. LX22NPO5103, Funded by the European Union–Next Generation EU), from the Czech Science Foundation (Grant No. 23-07021), and from US National Institutes of Health (National Institute for Allergy and Infectious Diseases, U19AI144177).

## Author contributions

Conceptualization: D.Š.; methodology: P.P., J.B., M.H. and E.V.; investigation: P.P., J.B. M.H., E.V. and L.K.; writing—original draft: D.Š., J.B., and P.P.; writing—review and editing: D.Š., J.B., P.P. and L.K.; funding acquisition: D.Š.; resources: D.Š. and L.K.; supervision: D.Š.

## Competing interests

The authors declare no competing interests.
