## [Transparent Peer Review file · Communications Medicine]

Resistance to ceftriaxone and penicillin G among contemporary syphilis strains confirmed by natural in vitro mutagenesis

Corresponding Author: Professor David Šmajs

Version 0:

Reviewer comments:

Reviewer #1

(Remarks to the Author)

This manuscript describes introduction of a point mutation into *T. pallidum* corresponding to a mutation observed in clinical isolates resistant to ceftriaxone and penicillin G. The work is important for at least two reasons. Methodologically, this is the first report using “natural” transformation to introduce an unmarked mutation in *T. pallidum*. Secondly, biological documentation of the resistance mechanism in vitro demonstrates the power of the recently-developed *T. pallidum* culture system. Introduction of A1873G into TP0705 (one of 5 annotated *T. pallidum* PBP’s) resulted at least partial resistance to ceftriaxone and penicillin G. Presumably further accumulation of mutations could eventually result in high level resistance. This has significant implications for the future of antibiotic treatment of syphilis and other treponematoses. The manuscript is well-written and rigorously documented, including appropriate statistical analysis. The only improvement I would suggest is to include the amino acid sequence change resulting from the A1873G change (assuming it does not introduce a stop codon).

Reviewer #2

(Remarks to the Author)

This is an interesting and relevant experimental study describing the use of in vitro mutagenesis to introduce mutations associated with ceftriaxone treatment failure into laboratory strains of *Treponema pallidum*. I have some comments for the authors to consider.

Comments:

1. Line 65-66: While some penicillin or ceftriaxone treatment failures may be due to resistance in *T. pallidum*, there is no documented clinical evidence of resistance. The authors should mention this here.
2. Line 66-67. The authors should provide some details on this treatment failure case. e.g., treatment dose, clinical outcome, etc.
3. Line 84-87: The background on PBP genes and SNPs should be moved to the introduction.
4. Line 150-151: Indicate which clade the isolates with alleles 4 and 6 fell into, if not the TPA clade.
5. Line 212: There are published papers showing that Nichols-like strains are more common than previously thought. Please cite relevant papers.
6. The authors should add a statement comparing their antibiotic susceptibility results to the in vitro susceptibilities described by Tantalo et al. 2023 (Ref #14).
7. Line 179-182: How does the experimental antibiotic concentrations used in this study translate to therapeutic levels for the authors to justify that B-lactam antibiotics can still be used for treatment?

Reviewer #3

(Remarks to the Author)

Review: COMMSMED-24-1027-T

In this manuscript, the authors used the *Treponema pallidum* in vitro co-culture model to evaluate the impact of a single point mutation on resistance to Penicillin G and Cephtriaxone. The manuscript includes two key innovations: to my knowledge, this is the first example of site directed mutagenesis for *T. pallidum*, and represents an exciting new tool for studying this pathogen. This study also provides the first experimental evidence of reduced susceptibility to Beta-lactams in syphilis, and this is an extremely important observation of great relevance to the STI field, particularly given the rapid increases in syphilis rates seen around the world in recent years.

Notably, Benzathine Penicillin (BPG – a different isomeric form to Penicillin G) is the recommended first-line treatment for syphilis, whilst neither of the drugs tested here is considered a frontline treatment for syphilis (Penicillin G is recommended for use only in neonates, and specifically not recommended for adults; Cephtriaxone is a 2nd line treatment in the European treatment guidelines after BPG and doxycycline). This point notwithstanding, the implications of reduced susceptibility to penicillins mediated by SNPs are of great clinical relevance.

Whilst the observation itself is significant, the manuscript seems preliminary and incomplete, with a number of areas of follow-up needed to firm up the observations and provide mechanistic and epidemiological support. It would benefit from further work to strengthen these areas.

1. One area that should be clarified is that a single point mutation (SNP) does not itself confer resistance/reduced susceptibility. It is the amino acid change and corresponding impact on the protein structure that has an effect. I found it surprising that the authors do not ever mention what the amino acid change is, and only ever refer to it as a nucleotide change – or is it a synonymous change (and thus has no clear functional impact?). Does the change alter an amino acid in an exposed loop or catalytic domain of the pbp – i.e. is it biologically plausible that this SNP would impact on susceptibility by changing secondary structure? This seems like a quick and simple thing to examine and would enhance the importance of the work. The authors should consider overlaying the SNP with e.g. AlphaFold structure predictions. As others have done previously (e.g. Sun 2016, Taouk 2022), they could also use predictive tools like sift to test the functional impact of the change on the pbp.
2. The authors' validation of their mutagenesis would benefit from further work. They used amplicon-based Sanger sequencing to confirm fixation of the A1873G allele. However, it is well established that Sanger sequencing is incapable of detecting low frequency minority variants below ~10%. Confirmation of fixation should be done using a more sensitive technique (e.g. deep amplicon sequencing).
3. Moreover, the authors have not confirmed that their mutagenesis experiment has not resulted in other changes in the genome – if possible, demonstrating this with whole genome sequencing of the mutant clone would strengthen this point.
4. As far as I can tell, the entire mutagenesis and AMR experiment represents a single biological replicate (one passage chain), and this has not been reproduced using a different strain, or even independently using the same strain. Whilst I appreciate that TP in vitro culture is very time consuming, providing replicate data would improve the manuscript. However, given the difficulties of the model, this may not be possible so the authors should at least comment on this limitation in their discussion.
5. I am also unclear how many technical replicates were used for the various experiments, e.g. antimicrobial susceptibility testing. Please clarify.
6. The authors have presented a streamlined abstraction of their processes and results in Figures 1 and 2. However, this means that there is limited detail of their passage runs (did TPA grow consistently over time and at what levels). This contrast with recent studies of the in vitro culture model examining selection using doxycycline (Tantalo 2024) which provided more comprehensive analysis of passage. Moreover, the authors choice to summarise their results in Figure 2 as a ratio makes it hard to interpret – greater clarity in the legend would be helpful, but I also suggest either using the raw data here, or providing it as a supplementary figure.
7. Figure 2 – the authors' use of so-called “dynamite” or “plunger” bar plots to describe continuous distribution data is unhelpful and misleading, since they hide the true distribution of the individual samples (see Drummond & Vowler, 2011, *Br J. Pharmacol* <https://www.ncbi.nlm.nih.gov/pmc/articles/PMC3087125/>). Moreover, error bars can also be very misleading, and in the case of Figure 2, the authors do not even state what those error bars represent (are they standard deviation, standard error, 95% confidence intervals, or something else?). The authors must replace bar/dynamite/plunger plots in their figures with a more appropriate figure type that adequately shows the distribution of sample values, e.g. boxplot, dotplot, or beehive plot.

Minor points

8. The authors discuss the disparity in population size between SS14-like and Nichols-like samples, suggesting that Nichols-like strains are “common amongst historical laboratory reference strains but rare among contemporary clinical isolates...”. The large number of TPA genomes published in recent years have demonstrated that this is not the case – Nichols-like strains are widely distributed in contemporary clinical populations (Beale 2021) and accounted for 23% of syphilis in a large recent Australian study (Taouk 2022) and 24% in the UK (Beale 2023). Therefore, ‘rare’ is not an appropriate term – ‘less common’ may be better. Moreover, whilst suggestive, a lineage being less common does not

necessarily imply evolutionary advantage or fitness (as the authors suggest on line 60) – in some cases the disparities may relate to ad hoc sampling biases (small number of samples from a single clinic which oversamples from a transmission network are used represent the TPA diversity of an entire country), and could also be a result of epidemiological factors, since some transmission networks may be larger and more mobile than others. I suggest moderating the language around this point.

9. Figure 2 – can the authors relate the experimental concentrations of cephtriaxone and penicillin used in vitro to clinical dosing regimens? Would current dosing with cephtriaxone be at risk, or can this not be evaluated?

10. The authors imply that the rise in syphilis rates may be linked to penicillin treatment failure, and proceeded to examine the prevalence of SNP using the PubMLST database. Previous authors (including those here) have suggested that the rise is also linked to macrolide resistance, and a very high percentage of *T. pallidum*, particularly some SS14 sublineages/subtypes are also macrolide resistance. Do the same strains/subtypes carry both macrolide and putative penicillin resistance mutations?

11. Line 201 – the authors state that “transformation mediated by natural competence appears to be more frequent than point mutations by at least several orders of magnitude...”. Please provide citations for this claim.

Reviewer #4

(Remarks to the Author)

Congratulations on this important study. There are some minor concerns such as referring to referring to 'syphilis bacteria' L 96 when one should refer to TPA.

Also I am not convinced that the Students t-test was appropriate for the analysis in Figure 2. Were the preconditions for this test met?

I cannot understand what is being demonstrated in Fig 2. In the text we are told to look to Fig 2 to see evidence that : "An increased resistance of the recombinant SS14 strain was statistically confirmed in all tested concentrations (Fig. 2). "L122

I do not see evidence of this in Fig 2. Where are the MICs. Also in the methods I do not see how MICs were evaluated. There is a nice description of how to do this using DNA copy numbers of the target gene described here for TPA:
[https://www.thelancet.com/journals/lanmic/article/PIIS2666-5247\(23\)00219-7/fulltext](https://www.thelancet.com/journals/lanmic/article/PIIS2666-5247(23)00219-7/fulltext)

Version 1:

Reviewer comments:

Reviewer #1

(Remarks to the Author)

The authors have addressed this reviewer's concerns.
I support publication of this manuscript as revised.

Reviewer #3

(Remarks to the Author)

In this revised manuscript, the authors have responded well to mine and the other reviewers' comments. The data is better presented, and the statistics more appropriate. The manuscript is greatly improved. I have a few remaining points.

Alphafold analysis – Thank you for adding this analysis. This new work is presented in the Discussion, but surely this is Results and Methods, and should therefore be described in those sections. Moreover, it is surprising that the M625V mutation has no identifiable impact on secondary structure. What is the putative mechanism for this mutation enabling reduced susceptibility? Is there any suggestion that the mutation is located near the active serine site in the transmembrane cleft of the PBP?

Discussion, Line 245 – based on prior data for macrolide resistance spread, the authors speculate about the emergence of penicillin resistance being decades away. However, the situations are not directly comparable – the authors' work suggests that the M625V mutation is common in SS14 strains and thus already widely distributed amongst global syphilis populations. Moreover, macrolide resistance occurred towards antimicrobials such as erythromycin and azithromycin, which have never been frontline antibiotics for syphilis. Indeed some authors (e.g. Beale 2019) have suggested that selection of macrolide resistance may have been linked to off-target treatment, rather than direct treatment of syphilis with macrolides. This further contrasts with penicillins, where Benzathine Penicillin is the first-line drug. Emerging penicillin resistance (e.g. due to accumulation of additional mutations that further reduce susceptibility) occurring in the right sexual network could lead to rapid global dissemination. I therefore think it is too strong to speculate that resistance is decades away – if it

emerges, it could become a problem much quicker.

Minor points:

A minor point, but whilst the manuscript is generally very well written, the new sections of text added (and tracked as changes) have some grammatical errors or poor English, and would likely benefit from further refinement.

Data availability – please ensure the genomic data (sequencing reads) are made available in an appropriate repository (e.g. SRA or ENA).

Reviewer #4

(Remarks to the Author)

I am happy that all my concerns have been addressed

Version 2:

Reviewer comments:

Reviewer #3

(Remarks to the Author)

The authors have comprehensively addressed any remaining concerns I had, and I am happy that this important study is ready for publication.

I note two very minor typos which should probably be addressed before publication:

- Line 140 - the authors misspell the tool "AlfaFold" instead of "AlphaFold".
- Line 213 - "..., suggesting its localization near the 213 penicillin-binding site." I am not quite clear what the authors are trying to say in this part after the brackets and this would benefit from rephrasing.

Mathew Beale

Referee expertise:

Referee #1: treponema biology, resistance

Referee #2: treponema, resistance

Referee #3: genomics, treponema, resistance

Referee #4: amr, genomics, treponema

Reviewers' comments:

Reviewer #1 (Remarks to the Author):

This manuscript describes introduction of a point mutation into *T. pallidum* corresponding to a mutation observed in clinical isolates resistant to ceftriaxone and penicillin G. The work is important for at least two reasons. Methodologically, this is the first report using "natural" transformation to introduce an unmarked mutation in *T. pallidum*. Secondly, biological documentation of the resistance mechanism in vitro demonstrates the power of the recently-developed *T. pallidum* culture system. Introduction of A1873G into TP0705 (one of 5 annotated *T. pallidum* PBP's) resulted at least partial resistance to ceftriaxone and penicillin G. Presumably further accumulation of mutations could eventually result in high level resistance. This has significant implications for the future of antibiotic treatment of syphilis and other treponematoses. The manuscript is well-written and rigorously documented, including appropriate statistical analysis. The only improvement I would suggest is to include the amino acid sequence change resulting from the A1873G change (assuming it does not introduce a stop codon).

We thank the reviewer for his comment. The corresponding amino acid sequence replacements have been added to the text on lines 91-95: „While sequences of loci TP0500, TP0547, TP0574, and TP0760 were found to be identical to the SS14 reference strain,²⁰ the TP0705 sequence revealed two nucleotide variants (A1873G and G2122A substitutions) compared to the SS14 strain. Both of these variants lead to amino acid change (M625V and G708S).“

Reviewer #2 (Remarks to the Author):

This is an interesting and relevant experimental study describing the use of in vitro mutagenesis to introduce mutations associated with ceftriaxone treatment failure into laboratory strains of *Treponema pallidum*. I have some comments for the authors to consider.

Comments:

1. Line 65-66: While some penicillin or ceftriaxone treatment failures may be due to resistance in *T. pallidum*, there is no documented clinical evidence of resistance. The authors should mention this here.

We have added this information in the sentence on lines 65-67: "However, reports on failed penicillin treatment are becoming more numerous⁹⁻¹¹ although there is no documented clinical evidence of resistance to penicillin due to laborious *in vitro* cultivation system for TPA."

2. Line 66-67. The authors should provide some details on this treatment failure case. e.g., treatment dose, clinical outcome, etc.

The details on the ceftriaxone treatment failure have been added to lines 85-90: „The three months old girl with serologically proven syphilis was treated with intravenous ceftriaxone therapy (100 mg/kg daily for 14 days). The follow-up serology showed RPR elevated titer five months later and buccal swab revealed TPA strain having identical allelic profile as before ceftriaxone therapy. The reinfection was excluded by testing for active infection in parents and treatment with intramuscular procain-benzylpenicillin followed (50,000 units/kg daily for 14 days)."

3. Line 84-87: The background on PBP genes and SNPs should be moved to the introduction.

The info on PBP genes and SNPs found in them has been added to lines of 69-71 of the Introduction section: „Genome of TPA codes for five penicillin-binding proteins (i.e., TP0500, TP0547, TP0574, TP0705, and TP0760 loci). Altogether 56 single nucleotide variants were described so far including 23 in TP0705.^{13,14}"

4. Line 150-151: Indicate which clade the isolates with alleles 4 and 6 fell into, if not the TPA clade.

The alleles 4 and 6 were found in two individual samples and the locus TP0705 was the only successfully determined sequence in these samples. Therefore, it was not possible to characterize these two samples with regard to TPA clade. This information has been added to lines 167-170: „While alleles 2, 5, 7, 11, 13, and 21 were found rarely (isolates of alleles 4 and 6 were not classified to TPA clade as these two alleles were found in two single isolates and TP0705 was the only locus determined), allele 1 was predominant (75.5%, 880 out of 1165 SS14-like isolates)."

5. Line 212: There are published papers showing that Nichols-like strains are more common than previously thought. Please cite relevant papers.

The analysis of typing studies which determined TPA clade and whole genome sequencing studies of clinical samples (published between 2006 and 2022) showed a massive prevalence of SS14 clade-belonging isolates (638 out of 4525 belonged to Nichols clade representing 14.1% of Nichols clade strains). We believe that this type of

analysis is more accurate than citing individual papers as individual studies may present biases (geographical location, inclusion of outbreaks) which is at least partially eliminated by pooling of all available data. However, this type of analysis would require citing all the available studies and thus the information regarding Nichols-like strains has been modified in the following manner (lines 261-269): „The partial resistance to penicillin in contemporary SS14-like strains of TPA could contribute to the paradox of Nichols-like strains^{5,40} which are being common among historical laboratory reference strains but less common among contemporary clinical isolates, especially in the developed countries. Recently, there are reports showing that Nichols-like strains are more common than analyzed previously^{41, 42}. Very recent report discovered almost 1:1 ratio between SS14 and Nichols clade in a set of 96 typeable clinical samples in Spain collected between years 2021 and 2023⁴³ and it will be certainly interesting to see how the prevalence of SS14 and Nichols clade isolates will be changing over time in different locations.”

6. The authors should add a statement comparing their antibiotic susceptibility results to the *in vitro* susceptibilities described by Tantaló et al. 2023 (Ref #14).

To address this point, we added the following paragraph to Results section (lines 149-154): “Moreover, in accordance with Tantaló,¹⁶ the secondary minimal inhibitory concentration (MIC) was determined for A1873G mutant strains. For ceftriaxone, both recombinant strains showed twice higher secondary MIC concentration compared to wild type strains (from 2.5 to 5.0 ng/ml). Similarly, both mutants showed higher secondary MIC of penicillin G. The MIC of DAL-1 mutant increased from 0.25 to 0.5 ng/ml, while trend for increased resistance was even higher for SS14 mutant (>0.5 ng/ml). For more details see Figure S2.”

7. Line 179-182: How does the experimental antibiotic concentrations used in this study translate to therapeutic levels for the authors to justify that B-lactam antibiotics can still be used for treatment?

Thank you for this interesting question. The following text has been added to the Discussion (lines 222-233): “The drug plasma concentration of ceftriaxone (29.7 mg/L¹⁶) seems to be sufficient to overcome the decrease in susceptibility of mutant TPA strain (0.0025 mg/L for SS14 strain and 0.005 mg/L for SS14 A1873G mutant). The drug plasma concentration of benzathine penicillin G (0.0125 mg/L¹⁶) should be also sufficient as secondary MIC assessed in our study for SS14 strain A1873G mutant is higher than 0.0005 mg/L which is at least 4-times higher than for the wild type strain. Limitation of this approximation includes the difference between benzathine penicillin G used for syphilis treatment and penicillin G potassium salt used in this study, but it was shown that relative growth inhibition of these two penicillin versions is similar *in vitro*¹⁷. However, tissue concentration of the drug may be different from plasma concentration and thus A1873G mutants should be tested in experimental rabbit infection to address whether therapeutic levels of penicillin are sufficient.”

Reviewer #3 (Remarks to the Author):

Review: COMMSMED-24-1027-T

In this manuscript, the authors used the *Treponema pallidum* in vitro co-culture model to evaluate the impact of a single point mutation on resistance to Penicillin G and Ceftriaxone. The manuscript includes two key innovations: to my knowledge, this is the first example of site directed mutagenesis for *T. pallidum*, and represents an exciting new tool for studying this pathogen. This study also provides the first experimental evidence of reduced susceptibility to Beta-lactams in syphilis, and this is an extremely important observation of great relevance to the STI field, particularly given the rapid increases in syphilis rates seen around the world in recent years.

Notably, Benzathine Penicillin (BPG – a different isomeric form to Penicillin G) is the recommended first-line treatment for syphilis, whilst neither of the drugs tested here is considered a frontline treatment for syphilis (Penicillin G is recommended for use only in neonates, and specifically not recommended for adults; Cephtriaxone is a 2nd line treatment in the European treatment guidelines after BPG and doxycycline). This point notwithstanding, the implications of reduced susceptibility to penicillins mediated by SNPs are of great clinical relevance.

Whilst the observation itself is significant, the manuscript seems preliminary and incomplete, with a number of areas of follow-up needed to firm up the observations and provide mechanistic and epidemiological support. It would benefit from further work to strengthen these areas.

1. One area that should be clarified is that a single point mutation (SNP) does not itself confer resistance/reduced susceptibility. It is the amino acid change and corresponding impact on the protein structure that has an effect. I found it surprising that the authors do not ever mention what the amino acid change is, and only ever refer to it as a nucleotide change – or is it a synonymous change (and thus has no clear functional impact?). Does the change alter an amino acid in an exposed loop or catalytic domain of the pbp – i.e. is it biologically plausible that this SNP would impact on susceptibility by changing secondary structure? This seems like a quick and simple thing to examine and would enhance the importance of the work. The authors should consider overlaying the SNP with e.g. AlphaFold structure predictions. As others have done previously (e.g. Sun 2016, Taouk 2022), they could also use predictive tools like sift to test the functional impact of the change on the pbp.

Thank you for this suggestion, the amino acid sequence changes have been added to the text on lines 94-95: "Both of these variants lead to amino acid change (M625V and

G708S).” Only one of these changes was introduced to the SS14 and DAL-1 strains (A1873G leading to M625V amino acid change). The AlphaFold structure prediction of the effect of the amino acid change is presented in the Supplementary Material as Supplementary Fig. S3. The corresponding commentary could be found in the Discussion on lines 211-215: „ We performed AlphaFold structure prediction^{25,26} of wild type penicillin binding protein TP0705 and the version with M625V amino acid change. The mentioned amino acid change is located in a α -helix and does not have major impact on secondary structure of the whole protein (Supplementary Fig. S3). Although we do not see any major differences, the impact on penicillin binding cannot be excluded..”

2. The authors’ validation of their mutagenesis would benefit from further work. They used amplicon-based Sanger sequencing to confirm fixation of the A1873G allele. However, it is well established that Sanger sequencing is incapable of detecting low frequency minority variants below ~10%. Confirmation of fixation should be done using a more sensitive technique (e.g. deep amplicon sequencing).

Thank you for this suggestion, we used Sanger sequencing for monitoring the process of mutation fixation analyzing the mutant cultures on weekly basis and we believe that concerns about fixation will be addressed in the following point where whole genome sequencing data are presented.

3. Moreover, the authors have not confirmed that their mutagenesis experiment has not resulted in other changes in the genome – if possible, demonstrating this with whole genome sequencing of the mutant clone would strengthen this point.

Thank you for this suggestion, we performed whole genome sequencing of two mutants produced in this study and results have been added to the Results section of the manuscript (lines 183-196): „ Whole genome sequencing of both recombinant strains obtained in this study was performed (SS14 strain carrying A1873G sequence change and DAL-1 strain carrying A1873G sequence change) and compared to wild type SS14 and DAL-1 strain sequences. Both strains were shown to carry A1873G sequence change in TP0705 (number of reads supporting this change was 923 out of 923 in SS14 recombinant strain and 1072 out of 1080 in DAL-1 recombinant strain). No other sequence changes common to both strains were detected. While DAL-1 recombinant strain harbours only A1873G sequence change, two additional SNPs were detected in SS14 recombinant strain, including G799578T in TP0732 (i.e., amino acid change T67K in bifunctional methylenetetrahydrofolate dehydrogenase (NADP(+))/methenyltetrahydrofolate cyclohydrolase) and C811984T in TP0746 (i.e., amino acid change A425T in pyruvate, phosphate dikinase) (genome coordinates and annotation based on CP004011). We concluded that it is unlikely that the observed effect in recombinant strains is caused by other nucleotide changes than the common sequence substitution introduced to TP0705.”

4. As far as I can tell, the entire mutagenesis and AMR experiment represents a single biological replicate (one passage chain), and this has not been reproduced using a

different strain, or even independently using the same strain. Whilst I appreciate that TP in vitro culture is very time consuming, providing replicate data would improve the manuscript. However, given the difficulties of the model, this may not be possible so the authors should at least comment on this limitation in their discussion.

The mutation was introduced into two different TPA strains. SS14 strain is presented in the body of the manuscript and DAL-1 strain is presented in the Supplementary material. This overcomes the concern of a single biological replicate. The following statement has been modified at the beginning of Discussion section (lines 199-202): „In this study, we showed that contemporary SS14-like strains of TPA have TP0705 alleles encoding partial resistance to ceftriaxone and penicillin G and perhaps other β -lactam antibiotics and that this resistance is encoded by the A1873G substitution which was experimentally introduced to SS14 and DAL-1 strains in an in vitro cultivation system.”

5. I am also unclear how many technical replicates were used for the various experiments, e.g. antimicrobial susceptibility testing. Please clarify.

This information is now specified on lines 367-372: „At least three biological replicates for each antibiotic concentration were performed and each of them was tested by qPCR analysis in duplicate. There were 10 biological replicates (20 values) for SS14 strain, 6 biological replicates (12 values) for recombinant SS14 A1873G strain, 5 biological replicates (10 values) for DAL-1 strain, and 3 biological replicates (6 values) for recombinant DAL-1 A1873G strain in ceftriaxone testing and 20, 10, 12 and 6 values in penicillin G testing, respectively.”

6. The authors have presented a streamlined abstraction of their processes and results in Figures 1 and 2. However, this means that there is limited detail of their passage runs (did TPA grow consistently over time and at what levels). This contrast with recent studies of the in vitro culture model examining selection using doxycycline (Tantalo 2024) which provided more comprehensive analysis of passage. Moreover, the authors choice to summarize their results in Figure 2 as a ratio makes it hard to interpret – greater clarity in the legend would be helpful, but I also suggest either using the raw data here, or providing it as a supplementary figure.

Thank you for this comment. The numbers of treponemes determined during selection of recombinant A1873G strains in both SS14 and DAL-1 were added to supplementary material (Supplementary Fig. S4). The process of isolation of the mutant was quite straightforward although very time consuming. The numbers of treponemes in culture were determined by dark field microscopy and each passage was examined by PCR to examine whether sequence of TP0705 contains wild type sequence or the mutant sequence (as described in Material and methods on lines 309-333). Once the mutation was fixed, we continued to monitor the presence of the recombinant version in the culture and also in each experiment of antibiotic testing where the recombinant version

of the sequence was confirmed. The whole genome sequencing of the recombinant SS14 A1873G and DAL-1 A1873G was performed as described under point 3 (reviewer 3). Thus, we do not include a detailed figure representing the process of isolation of the recombinant strains in the main body of manuscript.

We have chosen to present the data as ratios since our results come from multiple biological replicates derived from subsequent passages from different inocula rather than more technical replicates derived from one inoculum (as presented by Tantalo et al. 2023, reference 16 in our manuscript). Therefore, the raw data show more variation because the inoculum for each biological replicate was different and antibiotic susceptibility testing reflects variation of the growth under *in vitro* conditions. However, to show the full set of the data, we have changed the column graphs into boxplots together with presenting all data points. Moreover, a non-parametric test was used for statistical analysis (see Fig 2 and Supplementary Fig 1). The raw data are now presented as Supplementary Fig S2. We are convinced these changes address reviewer's concerns about the clarity of the data interpretation.

7. Figure 2 – the authors' use of so-called "dynamite" or "plunger" bar plots to describe continuous distribution data is unhelpful and misleading, since they hide the true distribution of the individual samples (see Drummond & Vowler, 2011, Br J. Pharmacol <https://www.ncbi.nlm.nih.gov/pmc/articles/PMC3087125/>). Moreover, error bars can also be very misleading, and in the case of Figure 2, the authors do not even state what those error bars represent (are they standard deviation, standard error, 95% confidence intervals, or something else?). The authors must replace bar/dynamite/plunger plots in their figures with a more appropriate figure type that adequately shows the distribution of sample values, e.g. boxplot, dotplot, or beehive plot.

Thank you for this comment. To show the full extent of the data, we have changed the column graphs into boxplots together with presenting all data points. Moreover, a non-parametric test was used for statistical analysis (see Fig 2 and Supplementary Fig 1). The raw data are now presented as Supplementary Fig S2. We are convinced these changes address reviewer's concerns about the clarity of the data interpretation.

The description of the Fig 2 includes the description of the graphical representation of the data (line 142-147): „Red line represents median value of the ratio, the boxplot shows IQR and color dots represent individual ratio values. At least three biological and two technical replicates were examined using qPCR and are shown. Motility data were obtained using dark-field microscopy. The Mann-Whitney test was used to calculate the statistical significance (*p < 0.05, **p < 0.01, and ***p < 0.001).”

Minor points

8. The authors discuss the disparity in population size between SS14-like and Nichols-like samples, suggesting that Nichols-like strains are "common amongst historical laboratory reference strains but rare among contemporary clinical isolates...". The large number of

TPA genomes published in recent years have demonstrated that this is not the case – Nichols-like strains are widely distributed in contemporary clinical populations (Beale 2021) and accounted for 23% of syphilis in a large recent Australian study (Taouk 2022) and 24% in the UK (Beale 2023). Therefore, 'rare' is not an appropriate term – 'less common' may be better. Moreover, whilst suggestive, a lineage being less common does not necessarily imply evolutionary advantage or fitness (as the authors suggest on line 60) – in some cases the disparities may relate to ad hoc sampling biases (small number of samples from a single clinic which oversamples from a transmission network are used to represent the TPA diversity of an entire country), and could also be a result of epidemiological factors, since some transmission networks may be larger and more mobile than others. I suggest moderating the language around this point.

Thank you for this suggestion. The information regarding Nichols-like strains has been modified in the following manner (lines 261-269): „The partial resistance to penicillin in contemporary SS14-like strains of TPA could contribute to the paradox of Nichols-like strains^{5,40} which are being common among historical laboratory reference strains but less common among contemporary clinical isolates, especially in the developed countries. Recently, there are reports showing that Nichols-like strains are more common than analyzed previously.^{41,42} Very recent report discovered almost 1:1 ratio between SS14 and Nichols clade in a set of 96 typeable clinical samples in Spain collected between years 2021 and 2023⁴³ and it will certainly be interesting to see how the prevalence of SS14 and Nichols clade isolates will be changing over time in different locations.“

9. Figure 2 – can the authors relate the experimental concentrations of cephtriaxone and penicillin used in vitro to clinical dosing regimens? Would current dosing with cephtriaxone be at risk, or can this not be evaluated?

Thank you for this interesting question. The following text has been added to the Discussion (lines 222-233): “The drug plasma concentration of ceftriaxone (29.7 mg/L¹⁶) seems to be sufficient to overcome the decrease in susceptibility of mutant TPA strain (0.0025 mg/L for SS14 strain and 0.005 mg/L for SS14 A1873G mutant). The drug plasma concentration of benzathine penicillin G (0.0125 mg/L¹⁶) should be also sufficient as secondary MIC assessed in our study for SS14 strain A1873G mutant is higher than 0.0005 mg/L which is at least 4-times higher than for the wild type strain. Limitation of this approximation includes the difference between benzathine penicillin G used for syphilis treatment and penicillin G potassium salt used in this study, but it was shown that relative growth inhibition of these two penicillin versions is similar in vitro¹⁷. However, tissue concentration of the drug may be different from plasma concentration and thus A1873G mutants should be tested in experimental rabbit infection to address whether therapeutic levels of penicillin are sufficient.“

10. The authors imply that the rise in syphilis rates may be linked to penicillin treatment

failure, and proceeded to examine the prevalence of SNP using the PubMLST database. Previous authors (including those here) have suggested that the rise is also linked to macrolide resistance, and a very high percentage of *T. pallidum*, particularly some SS14 sublineages/subtypes are also macrolide resistance. Do the same strains/subtypes carry both macrolide and putative penicillin resistance mutations?

Thank you for this question. Based on your suggestion we analyzed the available data on prevalence of A1873G sequence change in clinical samples using data available in the PubMLST database. At the same time, we do not believe that any authors including us suggest that macrolide resistance is directly linked to the recent rise of syphilis cases as many factors could contribute to this rise. Macrolide resistance phenotype was definitely linked to treatment failures and spread of syphilis in previous era when azithromycin was used as alternative treatment of contacts of syphilis cases due to its advantage of single dose oral regimen. Macrolides are no longer recommended for treatment of syphilis for many years as frequency of macrolide resistance reached very high levels. Moreover, it was shown that macrolide resistance emerges independently in different lineages (Beale et al. 2019) probably by the pressure of macrolide usage to treat other infections in the population. We believe that there are no data available on macrolide resistance phenotype and the corresponding fitness or infectivity of the isolate which could potentially lead to the rise of syphilis cases.

We believe that partial resistance to penicillin and ceftriaxone represents a selective advantage which could subsequently result in expansion of the corresponding sequence among contemporary circulating syphilis strains. This is exactly the evidence that we provide in this manuscript by the analysis of the prevalence of TP0705 A1873G variant among samples registered in the PubMLST database. We show that this variant is already very prevalent among circulating strains (Fig. 3). At the same time, we believe that macrolide resistance and resistance to penicillin and ceftriaxone are independent events induced by different antibiotic substances and their emergence in population had also different timelines. We believe that the information extracted from the PubMLST database that "only 95 out of 830 (11.4%) samples bearing allele 1 in the TP0705 are macrolide sensitive" provides information that both of these features are very prevalent among contemporary circulating strains. This suggests that antibiotic pressure by both antibiotic groups was significant in the past. Thus, we decided not to address this point in the manuscript.

11. Line 201 – the authors state that "transformation mediated by natural competence appears to be more frequent than point mutations by at least several orders of magnitude...". Please provide citations for this claim.

Thank you for this question. No citation can be provided for the mentioned statement as the conclusion is based solely on the comparison of the mutagenesis used in our study and the previously determined substitution rate in *Treponema pallidum*. The low

treponemal evolution rate (estimated to $2.8\text{--}4.1 \times 10^{-10}$ nucleotide substitutions per site per generation)^{38,39} is properly cited on line 251. The fact that we have selected the recombinant mutant in a culture with no more than 10^7 of cells leads us to conclusion that the event of transformation and recombination during our newly developed natural competence protocol is more frequent than the occurrence of point mutations by at least three ("several") orders of magnitude.

Reviewer #4 (Remarks to the Author):

Congratulations on this important study. There are some minor concerns such as referring to referring to 'syphilis bacteria' L 96 when one should refer to TPA.

Thank you for this note, syphilis bacteria has been changed to TPA (line 102).

Also I am not convinced that the Students t-test was appropriate for the analysis in Figure 2. Were the preconditions for this test met?

Thank you for this suggestion, we have used a non-parametric Mann-Whitney test as the data did not show normal distribution. The change from parametric to non-parametric test did not change the statistical support for the determined differences in antibiotic susceptibility.

I cannot understand what is being demonstrated in Fig 2. In the text we are told to look to Flg 2 to see evidence that : "An increased resistance of the recombinant SS14 strain was statistically confirmed in all tested concentrations (Fig. 2). "L122. I do not see evidence of this in Fig 2. Where are the MICs. Also in the methods I do not see how MICs were evaluated. There is a nice description of how to do this using DNA copy numbers of the target gene described here for

TPA: [https://www.thelancet.com/journals/lanmic/article/PIIS2666-5247\(23\)00219-7/fulltext](https://www.thelancet.com/journals/lanmic/article/PIIS2666-5247(23)00219-7/fulltext)

Thank you for this comment. We have chosen to present the data as ratios since our results come from multiple biological replicates derived from subsequent passages from different inocula rather than more technical replicates derived from one inoculum (as presented by Tantaló et al. 2023). Therefore, the raw data show more variation because the inoculum for each biological replicate was different and antibiotic susceptibility testing reflects variation of the growth under *in vitro* conditions. However, to show the full set of the data, we have changed the column graphs into boxplots together with presenting all data points. Moreover, a non-parametric test was used for statistical analysis (see Figure 2 and Supplementary Figure 1). The raw data are now presented as supplementary figure S2. We are convinced these changes address reviewer's concerns about the clarity of the data interpretation. To address the absence of MIC in the original version of the manuscript, we added the following paragraph to Results section (lines 149-154): "Moreover, in accordance with Tantaló,¹⁶ the secondary

minimal inhibitory concentration (MIC) was determined for A1873G mutant strains. For ceftriaxone, both recombinant strains showed twice higher secondary MIC concentration compared to wild type strains (from 2.5 to 5.0 ng/ml). Similarly, both mutants showed higher secondary MIC of penicillin G. The MIC of DAL-1 mutant increased from 0.25 to 0.5 ng/ml, while trend for increased resistance was even higher for SS14 mutant (>0.5 ng/ml). For more details see Figure S2."

Both approaches in data analysis (ratios and secondary MIC) show significant difference between recombinant strains and the corresponding wild type strains.

Reviewers' comments:

Reviewer #1 (Remarks to the Author):

The authors have addressed this reviewer's concerns.

I support publication of this manuscript as revised.

Thank you for your kind comment.

Reviewer #2 (Remarks to the Author): Satisfied.

Thank you for your kind comment.

Reviewer #3 (Remarks to the Author):

In this revised manuscript, the authors have responded well to mine and the other reviewers' comments. The data is better presented, and the statistics more appropriate. The manuscript is greatly improved. I have a few remaining points.

Alphafold analysis – Thank you for adding this analysis. This new work is presented in the Discussion, but surely this is Results and Methods, and should therefore be described in those sections. Moreover, it is surprising that the M625V mutation has no identifiable impact on secondary structure. What is the putative mechanism for this mutation enabling reduced susceptibility? Is there any suggestion that the mutation is located near the active serine site in the transmembrane cleft of the PBP?

We added Alphafold analysis to Methods section (lines 353-358) and Results section (lines 140-142). We do not intend to speculate about the putative mechanism of impact of this mutation on increased level of resistance as the similarity of amino acid sequence of TP0705 to penicillin binding proteins with known tertiary structure is rather low. Therefore we added the following statement commenting probability of the location of M625V mutation (lines 211-214): „The mentioned mutation resulted in the amino acid change M625V, located within an α -helix, and did not significantly alter the secondary structure of the whole protein (AlphaFold structure prediction,^{25,26} Supplementary Fig. S3), suggesting its localization near the penicillin-binding site.”

Discussion, Line 245 – based on prior data for macrolide resistance spread, the authors speculate about the emergence of penicillin resistance being decades away. However, the situations are not directly comparable – the authors' work suggests that the M625V mutation is common in SS14 strains and thus already widely distributed amongst global syphilis populations. Moreover, macrolide resistance occurred towards antimicrobials such as erythromycin and azithromycin, which have never been frontline antibiotics for syphilis. Indeed some authors (e.g. Beale 2019) have suggested that selection of macrolide resistance may have been linked to off-target treatment, rather than direct treatment of syphilis with macrolides. This further contrasts with penicillins, where Benzathine Penicillin is the first-line drug. Emerging penicillin resistance (e.g. due to accumulation of additional mutations that further reduce susceptibility) occurring in the

right sexual network could lead to rapid global dissemination. I therefore think it is too strong to speculate that resistance is decades away – if it emerges, it could become a problem much quicker.

We agree that situation of macrolide and penicillin resistance are not directly comparable. Thus, we replaced this section by milder description of possible outcome (lines 240-243): „These findings suggest that the emergence and spread of antibiotic resistance in TPA is a relatively common process. We, therefore, speculate that the emergence of clinically relevant penicillin resistance in syphilis is possible and could occur in a relatively near future.“

Minor points:

A minor point, but whilst the manuscript is generally very well written, the new sections of text added (and tracked as changes) have some grammatical errors or poor English, and would likely benefit from further refinement.

We carefully refined the text which was modified throughout both revisions of the manuscript and believe that English has improved significantly in the revised version „R2“.

Data availability – please ensure the genomic data (sequencing reads) are made available in an appropriate repository (e.g. SRA or ENA).

We have submitted sequencing reads to NCBI Bioproject database as stated on lines 348-351: „Sequencing data were submitted to the NCBI BioProject database under the following accession numbers PRJNA1232603 (SS14 strain carrying A1873G) and PRJNA1232606 (DAL-1 strain carrying A1873G).“

Reviewer #4 (Remarks to the Author): I am happy that all my concerns have been addressed

Thank you for your kind comment.

RESPONSE TO REVIEWERS' COMMENTS:

Reviewer #3 (Remarks to the Author):

The authors have comprehensively addressed any remaining concerns I had, and I am happy that this important study is ready for publication. I note two very minor typos which should probably be addressed before publication:

- Line 140 - the authors misspell the tool "AlfaFold" instead of "AlphaFold".

The typo was corrected (now line 276).

- Line 213 - "..., suggesting its localization near the 213 penicillin-binding site." I am not quite clear what the authors are trying to say in this part after the brackets and this would benefit from rephrasing.

The part after the brackets was modified to „suggesting that M625V replacement could modify the interaction with the penicillin molecule.” (lines 325-326)